# Piperine: Chemistry and Biology

**DOI:** 10.3390/toxins15120696

**Published:** 2023-12-12

**Authors:** Jin Han, Shaoyong Zhang, Jun He, Tianze Li

**Affiliations:** 1School of Public Administration, Xi’an University of Finance and Economics, Xi’an 710061, China; hanjin202012@126.com; 2Key Laboratory of Vector Biology and Pathogen Control of Zhejiang Province, College of Life Science, Huzhou University, Huzhou 313000, China; 1zhangshaoyong@163.com; 3College of Plant Protection, Northwest A&F University, Xianyang 712100, China; hejun6206@126.com

**Keywords:** piperine, piperine derivatives, biological activity, mechanism of action, structural modification, structure-activity relationship

## Abstract

Piperine is a plant-derived promising piperamide candidate isolated from the black pepper (*Piper nigrum* L.). In the last few years, this natural botanical product and its derivatives have aroused much attention for their comprehensive biological activities, including not only medical but also agricultural bioactivities. In order to achieve sustainable development and improve survival conditions, looking for environmentally friendly pesticides with low toxicity and residue is an extremely urgent challenge. Fortunately, plant-derived pesticides are rising like a shining star, guiding us in the direction of development in pesticidal research. In the present review, the recent progress in the biological activities, mechanisms of action, and structural modifications of piperine and its derivatives from 2020 to 2023 are summarized. The structure-activity relationships were analyzed in order to pave the way for future development and utilization of piperine and its derivatives as potent drugs and pesticides for improving the local economic development.

## 1. Introduction

Natural alkaloids are a huge library of promising lead compounds, exhibiting a variety of biological activities, which can provide multiple strategies for the development of drugs and pesticides. For instance, camptothecin, a well-known antitumor agent first isolated from *Camptotheca acuminata* Decne. in the mid-1960s, belongs to quinoline alkaloids [1,2]. The botanical products were utilized as pesticides to prevent pests in ancient China as early as the 7–5th centuries BC [3]. Nowadays, industrial chemical pesticides still occupy the dominant position of pesticides, but their high toxicity and environmental pollution have aroused many controversies among the public worldwide. Thus, plant secondary metabolites have become a major focus of research due to their potential application in developing new pesticides. Since 1985, neem-based insecticide Margosan-O, the first commercial pesticide with azadirachtin as the main ingredient, has been approved for registration in the United States, which triggered global enthusiasm for research on natural-based products from plants [4,5]. Plant-derived pesticides usually have the characteristics that their constituents are formed over the long period evolution process in nature and possess an integrated degradation mechanism. Due to their unique properties, they are difficult to accumulate and do not easily harm crops or promote resistance. It is estimated that more than 400,000 species of bioactive compounds in plants have been found, nevertheless, only 10% of the plant species have been investigated. Therefore, plant-derived products have a wide application prospect in the field of agriculture for the management of pests [6].

Black pepper (*Piper nigrum* L.), a family member of Piperaceae, is considered one of the most valuable spices and a potential candidate in natural product research, widely cultivated in lots of tropical and subtropical districts [7,8]. In addition, *P. nigrum* is acknowledged as the king of spices and consists of diverse bioactive compounds, such as sterols, fatty acids, terpenes, amides, and other phytoconstituents with the characteristics of various biological activities [9]. Piperine (**1**, Figure 1) is one of the major alkaloids isolated from the black pepper, which also occurs in the ripe fruits of *Piper longum* L. and the roots of *Piper sarmentosum* Roxb. [10,11]. The pepper processing industry participates in a huge number of various wastes that contain lots of piperine. To effectively utilize the by-products, people investigated the conditions of pressure and temperature on the method of supercritical CO_2_ (SC-CO_2_) extraction for two steps (100 bar, 60 °C for 1 h and 300 bar, 60 °C for 2 h) to obtain piperamide piperine in a gentle and efficient way [12]. A recent study reported that PFPE-CH, a dietary supplement made by mixing a low piperine fractional *Piper nigrum* extract (PFPE) with cold-pressed coconut oil and honey in distilled water, can help decrease the risk of tumor formation and reduce the side effects of chemotherapeutic drugs during breast cancer treatment. This is promising news for those undergoing treatment for breast cancer [13]. In the early 1970s, a piperine analogue Ilepcimide was originally extracted from a Chinese folk remedy, which was discovered as a safe and effective antiepileptic drug through structural characterization. In addition, the hospital of Peking University (Beijing, China) has been clinically implemented in more than 100,000 patients, and the results showed that the effective rate of Ilepcimide was 95.6% [14,15]. By 2020, the actual cultivated area of pepper in Yunnan Province had reached 39.4 square kilometers, the yield was 1070.8 tons, and the output values had achieved 42.8 million RMB [16]. The piperine molecular contains a dioxymethylene ring, an aliphatic olefin chain, and an amide group. Piperine and its derivatives possess many biological activities, including anticancer [17], antitumor [18], antiviral [19], anti-inflammatory [20], antimalarial [21], antioxidant [22,23], anti-Alzheimer’s disease [24,25,26], and anti-Parkinson disease properties [27]. The characteristics of the structure of piperine allow for modification at multiple sites so as to obtain more candidates with better activities, which are widely used in the agricultural field. For instance, Zhang et al. synthesized a series of bisamide-type piperine derivatives and evaluated them for insecticidal activity against *Plutella xylostella* (L.). The study results revealed that most of the desired compounds displayed better insecticidal activity compared to piperine. In particular, the molecular docking results indicated that a synthetic compound could act on γ-aminobutyric acid receptors [28]. Recently, our studies synthesized two novel series of ester and oxime ester piperine-type derivatives, some compounds displayed more than 100 times the acaricidal activity of piperine against *Tetranychus cinnabarinus* (Bois.). Moreover, the scanning electron microscope (SEM) demonstrated that their potential acaricidal activities may be related to the destruction of the construction of the cuticle layer crest of *T. cinnabarinus* [29,30]. Wang et al. found that the ester derivative prepared from piperine exhibited particularly significant broad-spectrum fungicidal activity against *Rhizoctonia solani* Kühn, *Fusarium graminearum* Schwabe, *Alternaria tenuis* Nees, *Gloeosporium theae-sinensis* I. Miyake, *Phytophthora capsica* Leonian, and *Phomopsis adianticola* [31].

Generally, with the characteristics of easy modification and lots of bioactivities, piperine and its derivatives have received much attention in the fields of synthetic organic chemistry and medicinal chemistry. In this review, many works concerning piperine and its derivatives have been conducted, we widely summarized the advances in the construction and structural modification of piperine and its derivatives from 2020 to 2023. Meanwhile, the latest biological activities, mechanisms of action, and structure-activity relationships of piperine and its derivatives are also presented.

## 2. Bioactivities of Piperine and Its Derivatives

### 2.1. Anticancer Activity

Cancer still poses the biggest danger to human health. For example, breast cancer is an increasing global health challenge. According to the World Health Organization (WHO), in 2020, approximately 2.3 million women were diagnosed with breast cancer and there were 685,000 deaths reported worldwide [32]. Rajarajan et al. reported that the dietary piperine represented a potential candidate that could suppress obesity-associated breast cancer growth and metastasis by regulating the miR-181c-3p/PPARα Axis [33]. Furthermore, Jeong et al. also found that piperine has anticancer properties such as growth inhibition, anti-migration, and anti-invasion of cancer cells [34]. In order to application of piperine derivatives as more potential anticancer drugs, structural modification has received much attention. For example, compound **2** (Figure 2) was evaluated for cytotoxicity and inhibition of TrxR (thioredoxin reductase) activity, paving the way for further exploration of piperine derivatives as TrxR inhibitors [35]. Elimam et al. examined the piperine-based sulfonamide **3** (Figure 2) for its anticancer activity towards the breast MCF-7 cancer cell line. The results indicated that it is a promising lead compound for developing efficient anticancer candidates with potent carbonic anhydrase (CA) inhibitory activity [36]. In addition, to validate the in silico results against triple-negative breast cancer (TNBC) MDA-MB-231, an in vitro vascular endothelial growth factor receptor-2 (VEGFR-2) inhibition assay was conducted. Compound **4** (Figure 2) exhibited powerful inhibitory activity with a half-maximal inhibitory concentration (IC_50_) value of 231 nM [37]. Pandya et al. found that piperine analogue **5** (Figure 2) may serve as an anticancer therapeutic seeing it affects c-myc oncogene expression via G-quadruplex mediated mechanism [38].

### 2.2. Antiviral Activity

The respiratory illness caused by the pandemic H1N1 influenza virus has become a significant global health concern. Mohammed et al. conducted a study in which they screened a piperine derivative **6** (Figure 3) for its antiviral activity and compared it with other strains in vitro on the MDCK cell line. The adsorption assay demonstrated a dose-dependent reduction of viral plaque with an EC_50_ value of 0.33 μM [39]. Over the past five years, the COVID-19 (SARS-CoV-2) epidemic has spread around the world, bringing immeasurable damage to people’s lives and health. However, some potential molecules showed promising activities against coronavirus, a published research illustrated that the plant alkaloid piperine could inhibit the vesicle fusion mediated by SARS-CoV-2 peptides and reduce the titer of SARS-CoV-2 progeny in vitro in Vero cells [40]. In addition, largemouth bass virus (LMBV) is a systemic viral pathogen that affects cultivated largemouth bass, causing high mortality rates. Wang et al. investigated the antiviral activity of piperine against LMBV in vitro and in vivo, and the results showed that piperine could act as a therapeutic and preventative drug for further study against LMBV infection [41].

### 2.3. Anti-Inflammatory Activity

Ultraviolet (UV) rays from sunlight are one of many environmental insults that harm the skin. Jaisin et al. discovered that piperine may effectively treat skin inflammation caused by UV irradiation [42]. Acute pancreatitis (AP) is a common acute abdominal disease characterized by pancreatic acinar cell death and inflammation. Huang et al. confirmed that piperine can target the endoplasmic reticulum autophagy (ER-phagy), providing a new insight into the pharmacological mechanism of piperine in treating AP [43]. Psoriasis is a common chronic inflammatory skin disease that is prone to relapse and difficult to cure. Studies have shown that piperine can help alleviate psoriasis by reducing epidermal hyperplasia, inflammatory cell infiltration, and the expression of psoriasis-characteristic cytokines, chemokines, and proteins in IMQ-induced psoriasiform dermatitis. Therefore, piperine has the potential to become an effective agent for psoriasis and may provide new strategies for clinical intervention [44]. Additionally, sciatica is a kind of combined pain caused by stimulation and compression of various factors, resulting in stabbing, burning, and dull pain along the route of the sciatic nerve and in the surrounding areas, which brings great physical and psychological pain to patients. Wang et al. confirmed the regulatory relationship between miR-520a and p65. In addition, they investigated the impact of miR-520a/P65 on cytokine levels following piperine stimulation to determine its therapeutic role in sciatica. As a result, they found that piperine can help alleviate pain. Specifically, piperine can promote the expression of miR-520a, which directly targets and inhibits the expression of P65, down-regulate pro-inflammatory factors IL-1β and TNF-α, while up-regulating the effects of anti-inflammatory factors IL-10 and TGF-β1. As a result, piperine may as a treatment for sciatica [45]. Duan et al. suggested that piperine may have anti-inflammatory properties by regulating the key factors of the NF-*κ*B and MAPK signaling pathways [46]. Yang et al. developed a novel piperine derivative **7** (Figure 3) and demonstrated that it exhibited potent therapeutic effects against neuroinflammation, which might be partly attributed to its inhibitory activity on kelch-like ECH-associated protein (Keap1)-nuclear factor erythroid-2-related factor 2 (Nrf2) protein-protein interaction [47]. Tian et al. synthesized a potent inhibitor **8** (Figure 3) of fatty acid amide hydrolase (FAAH) with an IC_50_ value of 0.65 μM. And the inhibitor was found to attenuate the lipopolysaccharide (LPS)-induced activation of BV2 cells, showing a significant anti-inflammatory activity [48].

### 2.4. Insecticidal Activity

In 2022, Oliveira et al. found that piperine loaded into nanostructured systems might be an effective drug to improve larvicidal activity against *Aedes aegypti* L. [49]. Yang et al. identified that compound **9** (Figure 3) was the most effective multichitinase inhibitor and exhibited higher insecticidal activity against *Ostrinia furnacalis* (Guenée) (Asian corn borer) than dual- or single-chitinase inhibitors. Results of molecular mechanism studies showed that compound **9** can interact with two conserved TRP and TYR of three chitinases in identical ways through hydrogen bonds, hydrophobic, and *π*-*π* interactions. Furthermore, the microinjection experiment revealed that the agent displayed significant sublethal effects against *O. furnacalis* by regulating its growth and development [50]. Zhang et al. synthesized a series of piperine derivates containing a linear bisamide. Among them, compound **10** (Figure 3) gave rise to 90% mortality at 1.0 mg/mL concentration against *P. xylostella* [28]. In order to investigate the effects of potential trehalase inhibitors in *Spodoptera frugiperda* (Smith), compounds **11** (Figure 3) and **12** (Figure 3) were synthesized by Zhong et al. The results showed that compound **12** significantly reduced trehalase activity. Additionally, compound **11** not only can reduce membraned-bound trehalase activity, but also inhibit the expression of *SfTRE2*, *SfCHS2*, and *SfCHT*, thus affecting the chitin metabolism, providing a theoretical basis for the application of trehalase inhibitors in the control of agricultural pests [51].

### 2.5. Antibacterial and Antifungal Activities

A series of piperine amide derivatives were synthesized by Sivashanmugam et al. Of all the derivates, compounds **13** (Figure 4) and **14** (Figure 4) worked exceptionally well against Gram-negative bacteria (*Escherichia coli* and *Acinetobacter baumannii*) and Gram-positive bacteria (*Staphylococcus aureus*, *Escherichia faecalis*, and *Staphylococcus epidermidis*) [52]. Das et al. found that piperine exhibited potent antibiofilm activity against *Pseudomonas aeruginosa* by accumulating reactive oxygen species, affecting cell surface hydrophobicity, and quorum sensing. This research suggested that piperine can effectively disrupt the biofilm formation of *P. aeruginosa*, offering a sustainable solution for protecting public health [53]. The emergence and rapid spread of multidrug-resistant (MDR) bacteria, such as *Vibrio cholerae*, is a major global public health concern. Piperine has been found to show a dose-dependent bactericidal effect on *V. cholerae* growth, regardless of their biotypes and serogroups at the concentrations of 200 and 300 μg/mL, respectively. It also can inhibit the growth of multidrug-resistant (MDR) strains of *P. aeruginosa* and *E. coli* isolated from poultry, and enterohemorrhagic/enteroaggregative *E. coli* O104 in 200 μg/mL. The results demonstrated that piperine has antimicrobial properties against pathogenic bacteria, including MDR strains, making it a potential therapeutic and preventative agent against infections [54]. Souza, Jr. et al. synthesized the compound **15** (Figure 4) and evaluated its antifungal activity against Candida, Trichophyton, and Microsporum strains. As a result, compound **15** exhibited 70% inhibition in seven tested strains, such as *Candida albicans* ATCC 76645, LM-111, LM-122 and *Candida krusei* LM-656, LM-13 and *Microsporum canis* LM-12, and *Microsporum gypseum* LM-512, with a minimum inhibitory concentration (MIC) ranging from 1.23–2.46 μmol/mL and a minimum fungicide concentration (MFC) ranging from 9.84–19.68 μmol/mL [55].

### 2.6. Other Activities

A series of N-aryl amide derivatives of piperine (**16**–**18**, Figure 4) were prepared by semi-synthesis, and these compounds were examined for their antitrypanosomal, antimalarial, and anti-SARS-CoV-2 main protease activities. Among them, compound **18** exhibited the most robust biological activities with no cytotoxicity against mammalian cell lines Vero and Vero E6, its IC_50_ values for antitrypanosomal activity against Trypanosoma brucei rhodesiense was 15.46 ± 3.09 μM, and its antimalarial activity against the 3D7 strain of Plasmodium falciparum was 24.55 ± 1.91 μM, which were 4-fold and 5-fold higher than that of piperine, respectively. Furthermore, compound **18** inhibited the activity of 3C-like main protease (3CL^Pro^) toward anti-SARS-CoV-2 activity with the IC_50_ value of 106.9 ± 1.2 μM. Docking and molecular dynamic simulation indicated that the potential binding of compound **18** in the 3CL^pro^ active site had improved binding interaction and stability [21]. Peroxisome proliferator-activated receptor γ (PPARγ) plays a key role in glucose, which is a ligand-mediated transcription factor. The lipid homeostasis always serves as a pharmacological target for new drug discovery and development. Wang et al. synthesized and evaluated some compounds for their agonistic activities of PPARγ. In particular, compound **19** (Figure 4) presented itself as a potential PPARγ agonist with an IC_50_ value of 2.43 μM, and the molecular docking studies indicated that compound **19** stably interacts with the amino acid residues of the PPARγ complex active site [56]. Piperine derivate **20** (Figure 4) containing benzodioxole molecule was identified as the promising antiparasitic candidate against Leishmania amazonensis [57]. Hsieh et al. investigated the protective effects of piperine on nerve growth factor (NGF) signaling in a kainic acid (KA) rat model of excitotoxicity. The results showed that piperine can protect hippocampal neurons against KA-induced excitotoxicity by enhancing the NGF/TrkA/Akt/GSK3β signaling pathways [58]. Alsareii et al. conducted an in vivo study and histopathological examination, which revealed early and intrinsic healing of wounds with the piperine-containing bioactive hydrogel system compared to the bioactive hydrogel system without piperine. These findings established that the piperine-containing bioactive hydrogel system is a promising therapeutic approach for wound healing applications [59].

## 3. Structural Modifications of Piperine and Its Derivatives

### 3.1. Structural Modifications at the Dioxymethylene Ring or the Aliphatic Olefin Chain of Piperine

As described in Figure 1 and Figure 2, Li et al. and Lv et al. regio- and stereo-selectively synthesized a series of piperine-type ester derivates **23a**–**23z** and **23a’**–**23l’**, and oxime ester derivates **25a**–**25z** and **25a’**–**25h’**, respectively, by using the Vilsmeier–Haack–Arnold (VHA) reaction. Meanwhile, as shown in Figure 3, by changing the substituent groups of dioxymethylene ring, compounds **33a**–**33i** were also prepared by Lv et al. All the synthetic compounds were evaluated for their acaricidal activities against T. cinnabarinus, among them, compounds **23e**, **23f**, **23u**, **23v**, **25f**, **25l**, **25u**, and **25v** showed significant acaricidal activities with the LC_50_ values ranging from 0.12 to 0.19 mg/mL (Table 1), which were comparable to that of the commercial acaricidal agent spirodiclofen (LC_50_: 0.12 mg/mL), and displayed almost 100-fold higher acaricidal activities than piperine (LC_50_: 14.20 mg/mL) [28,29,60]. As shown in Figure 4, Gu et al. used the catalyst **RhH-1** to reduce the aliphatic olefin chain of piperine under a mild condition with a high yield to obtain analogue **34** [61].

As illustrated in Figure 5, Luo et al. synthesized four piperine derivates **35** (4R,5R), **36** (4S,5S), **37** (4S,5R), and **38** (4R,5S), which were established by NMR, optical rotation, and CD spectra [62]. As demonstrated in Figure 6, the sample amine groups were introduced at the aliphatic olefin chain of piperine through a trisulfur-radical-anion-triggered C(sp^2^)-H amination of α,β-unsaturated carbonyl to obtain derivate **39** [63]. As presented in Figure 7, Wojtowicz-Rajchel et al. synthesized five novel corresponding cycloadducts **41**–**45** in moderate yields by the reaction of prochiral N-alkyltrifluoromethyl-methylene nitrones with piperine under non-catalyzed condition [64].

### 3.2. Structural Modifications at the Amide Group of Piperine

As shown in Figure 8 and Table 2, Tantawy et al. designed and synthesized a series of piperine-based dienehydrazide derivatives **49a**–**49u** and evaluated their insecticidal activities against third-instar larval of Culex pipiens L. Among all synthetic derivates, compounds **49a**, **49b**, **49f**, **49g**, **49m**, **49n**, **49o**, **49p**, and **49u** displayed better activities than piperine and deltamethrin (a commercial positive control). Molecular modeling revealed several interactions between derivates and acetylcholinesterase (AChE) substrate binding sites responsible for binding and inhibition [65]. As shown in Figure 9, twenty-two piperine derivates **2**, **52a**–**52l**, and **53a**–**53i** were synthesized by Zhong et al. and evaluated for their anticancer properties [34]. As shown in Figure 10, a series of novel piperine derivates **56a**–**56z**, **56a’**–**56b’**, and **59a**–**59b** containing a linear bisamide were synthesized and evaluated for their insecticidal activities against *P. xylostella* [50].

Human Cytochrome P450 2J2 (CYP2J2) plays an important role in metabolizing polyunsaturated fatty acids (PUFAs). As described in Figure 11, a series of piperine derivatives **62a**–**62k** were designed and synthesized based on the underlying interactions of piperine with CYP2J2. Among them, compounds **62j** and **62k** were developed as much stronger inhibitors and their inhibition activities increased approximately 10-fold higher than piperine with the IC_50_ values of 40 and 50 nM, respectively (Table 3) [66].

## 4. Mechanisms of Action of Piperine and Its Derivatives

The natural product piperine (**1**) and its derivates exhibit wide biological activity in the fields of medicine and agriculture. For example, piperine and its derivates have anticancer properties such as growth-inhibition, anti-migration, and anti-invasion of cancer cells by regulating the miR-181c-3p/PPARα Axis [32] and inhibiting thioredoxin reductase and carbonic anhydrase [34,35]. In addition, piperine can target endoplasmic reticulum autophagy (ER-phagy) in treating acute pancreatitis [42]. On the other hand, the insect chitinase OfChtI from the agricultural pest O. furnacalis is a promising target for green insecticide design. Han et al. first found that piperine can inhibit the insect chitinase from O. furnacalis. Piperine derivates **63a**–**63f** (Figure 5) were designed and synthesized by introducing a butenolide scaffold into the lead compound piperine. The results of the enzymatic activity assay revealed that the synthetic compounds (K_i_ = 1.03–2.04 μM) were approximately 40–80 times more effective in inhibiting OfChtI than the lead compound piperine (K_i_ = 81.45 μM). The inhibitory mechanism demonstrated that the introduced butenolide skeleton improved the binding affinity to OfChtI [67].

## 5. Structure-Activity Relationships of Piperine and Its Derivatives

The SARs analysis of piperine and its derivatives are described in Figure 6, and the detailed descriptions are as follows: (i) introduction of hydroxyl groups at C-8 and C-9 positions can obtain a potent antiviral candidate [38]. Replacing the dioxymethylene ring with a 2-oxazoline heterocyclic ring and introducing a Cl atom at the C-10 position can produce an anti-inflammatory agent, which has potent therapeutic effects against neuroinflammation [46]; (ii) introduction of ester and oxime ester groups at C-2 position of piperine is important for improving acaricidal activity. In particular, the long aliphatic chain esters or oxime esters, and a very crucial prerequisite is to retain the dioxymethylene ring [28,29,60]; (iii) introduction of the dienehydrazide groups can obtain potent insecticidal compounds against third-instar larval of *C. pipiens* [65]. The introduction of different substituted amides is beneficial for anticancer activity and can obtain a potential inhibitor against CYP2J2 [34,67]. Piperine derivates containing a linear bisamide exhibited significant insecticidal activity against *P. xylostella*, and they can be further studied as lead pesticidal agents [50].

## 6. Conclusions

In recent years, research and development of plant-derived pesticides in China have been steadily improving. The commercialization of plant-derived pesticides should capitalize on their benefits and prioritize sustainability. In the future, plant-derived pesticides will play a significant role in food security and the agricultural field for improving the local economic development. This review has summarized the recent developments from 2020 to 2023 on biological activities, structural modifications, and structure-activity relationships of piperine and its derivatives, in addition to their mechanisms of action. The structural characteristics of piperine implied that it had various sites for structural modifications, making it a high-value-added lead compound. It is prospective that this paper can provide essential information and proposals for further design and development of novel piperine derivatives as potent natural-product-based drugs and pesticides for improving the local economic development in the future.

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
