# Peer review of "Piperine: Chemistry and Biology"

_toxins, 2023, doi:10.3390/toxins15120696_

Round 1

Reviewer 1 Report

Comments and Suggestions for Authors

The manuscript the potential use of piperine as a plant-derived pesticide. While the manuscript is generally well-written, there may be potential for reader confusion. The abstract and conclusion section emphasizes its agricultural bioactivity, particularly as an environmentally friendly pesticide. In addition, the mechanism of piperine pesticides were written in the section “4. Mechanisms of Action of Piperine and Its Derivatives”, and “5. Structure-Activity Relationships of Piperine and Its Derivatives”. However, the section titled "2. Bioactivities of Piperine and Its Derivatives" delves into medical applications of piperine, including its anti-cancer, anti-viral, and anti-inflammatory properties. To reader understanding, the topic of this manuscript should restructure in the field of agriculture or medicine.

Author Response

Thanks for your great advice. In constructing this review of piperine, we found that the agricultural activities of piperine have only been reported in some literature in recent years. Thus, we generalized the medical activities of piperine, and also sorted out the agricultural activities in sections ‘2.4. Insecticidal Activity’ and ‘2.5. Antibacterial and Antifungal Activities’, respectively. If there are recent reports on the agricultural activity of piperine, we will summarize them in this manuscript. Thanks again!

Reviewer 2 Report

Comments and Suggestions for Authors

Dear Authors,

Your manuscript is well-written and interesting. The references are suitable. I have particularly tried to improve the binomial nomenclature and have corrected a few errors.

I have attached the manuscript with my comments.

Author Response

Thank you very much for your corrections. We have carefully corrected the errors in the manuscript and recorded the binomial nomenclature to avoid such ambiguity. Thanks again!

Reviewer 3 Report

Comments and Suggestions for Authors

Dear Authors, 

The manuscript contains very good information. The only concern is the language. I will suggest a major revision of the language.

Please see the pdf file for more corrections

Comments on the Quality of English Language

There is the requirement of a major language editing. 

Thank You

Author Response

Thank you very much for your corrections. We have carefully corrected some errors, and re-frame the sentences in the manuscript. In future writing, we will be more attentive to these issues. Thanks again!